

# Psychometric evaluation of an Italian custom 4-item short form of the PROMIS anxiety item bank in immune-mediated inflammatory diseases: an item response theory analysis

Marco Tullio Liuzza[1,*], Rocco Spagnuolo[2,*], Gabriella Antonucci[3,4], Rosa Daniela Grembiale[5], Cristina Cosco[2], Francesco Salvatore Iaquinta[5], Vanessa Funari[5], Stefano Dastoli[5], Steven Nistico[5] and Patrizia Doldo[2]

[1] Department of Medical and Surgical Sciences, "Magna Graecia" University of Catanzaro, Catanzaro, Calabria, Italy
[2] Department of Clinical and Experimental Medicine, "Magna Graecia" University of Catanzaro, Catanzaro, Calabria, Italy
[3] IRCCS Santa Lucia Foundation, Rome, Lazio, Italy
[4] Department of Psychology, University of Roma "La Sapienza", Rome, Lazio, Italy
[5] Department of Health Sciences, "Magna Graecia" University of Catanzaro, Catanzaro, Calabria, Italy
* These authors contributed equally to this work.

Corresponding authors
Marco Tullio Liuzza, liuzza@unicz.it
Rocco Spagnuolo, spagnuolo@unicz.it

## ABSTRACT

**Background:** There has recently been growing interest in the roles of inflammation in contributing to the development of anxiety in people with immune-mediated inflammatory diseases (IMID). Patient-reported outcome measures can facilitate the assessment of physical and psychological functioning. The National Institutes of Health (NIH)'s Patient-Reported Outcomes Measurement Information System (PROMIS®) is a set of Patient-Reported Outcomes (PROs) that cover physical appearance, mental health, and social health. The PROMIS has been built through an Item Response Theory approach (IRT), a model-based measurement in which trait level estimates depend on both persons' responses and on the properties of the items that were administered. The aim of this study is to test the psychometric properties of an Italian custom four-item Short Form of the PROMIS Anxiety item bank in a cohort of outpatients with IMIDs.

**Methods:** We selected four items from the Italian standard Short Form Anxiety 8a and administered them to consecutive outpatients affected by Inflammatory Bowel disease ($n = 246$), rheumatological ($n = 100$) and dermatological ($n = 43$) diseases, and healthy volunteers ($n = 280$). Data was analyzed through an Item Response Theory (IRT) analysis in order to evaluate the psychometric properties of the Italian adaptation of the PROMIS anxiety short form.

**Results:** Taken together, Confirmatory Factor Analysis and Exploratory Factor analysis suggest that the unidimensionality assumption of the instrument holds. The instrument has excellent reliability from a Classical Theory of Test (CTT) standpoint (Cronbach's α = 0.93, McDonald's ω = 0.92). The 2PL Graded Response Model (GRM) model provided showed a better goodness of fit as compared to the
1PL GRM model, and local independence assumption appears to be met overall. We did not find signs of differential item functioning (DIF) for age and gender, but evidence for uniform (but not non-uniform) DIF was found in three out of four items for the patient *vs.* control group. Analysis of the test reliability curve suggested that the instrument is most reliable for higher levels of the latent trait of anxiety. The groups of patients exhibited higher levels of anxiety as compared to the control group ($p$s < 0.001, Bonferroni-corrected). The groups of patients were not different between themselves ($p = 1$, Bonferroni-corrected). T-scores based on estimated latent trait and raw scores were highly correlated (Pearson's $r = 0.98$) and led to similar results.

**Discussion:** The Italian custom four-item short form from the PROMIS anxiety form 8a shows acceptable psychometric properties both from a CTT and an IRT standpoint. The Test Reliability Curve shows that this instrument is mostly informative for people with higher levels of anxiety, making it particularly suitable for clinical populations such as IMID patients.

## INTRODUCTION

The prevalence of anxiety is three times greater in patients with chronic diseases than in the general population (*Abbott et al., 2015*). In such patients, presence of these psychological disorders can be associated with an increase in mortality, in morbidity (*Ellis, Zhao & Egede, 2010*; *Nicholson, Kuper & Hemingway, 2006*), and, importantly, a deterioration in quality of life (*Dickens et al., 2008*; *Dickens et al., 2004*; *Moussavi et al., 2007*) with a consequent increase in the costs of healthcare. Disease and treatment factors are related to the determination of anxiety and depression, such as negative beliefs about the disease, the presence of pain, disability, and unpleasant side effects of treatment (*Dickens et al., 2008*; *Harrison & Maguire, 1994*).

There has recently been growing interest in the interplay between inflammation and mood disorders. Potential mechanisms involved include direct effects of pro-inflammatory cytokines monoamine levels, dysregulation of the hypothalamic–pituitary–adrenal (HPA) axis, pathologic microglial cell activation, impaired neuroplasticity and structural and functional brain changes (*Rosenblat et al., 2014*). Moreover, current evidence have shown that mood disorders are expected from higher levels of inflammatory mediators such as IL-6 and TNF-α (*Himmerich et al., 2008*; *Howren, Lamkin & Suls, 2009*; *Schmidt, Kirkby & Himmerich, 2014*) and have shown improved outcomes in patients when anti-inflammatory agents such as Acetyl-salicylic acid (ASA), celecoxib, anti-TNF-αagents, minocycline,curcumin and omega-3 fatty acid are used as an adjunct to conventional therapy. Inflammation contributes to the development of anxiety in people with immune-mediated inflammatory diseases (IMIDs), a group of ostensibly unrelated

conditions that share common inflammatory pathways encompasses about 80 diseases, including inflammatory bowel disease (IBD), Rheumatologic, and dermatological disorder (*Kuek, Hazleman & Östör, 2007*).

A recent systematic review and meta-analysis of randomized controlled trials regarding the effects of TNF-α inhibitor therapy on depression and anxiety in people with chronic physical illness, has shown that out of a total of 2,540 patients affected by rheumatic and dermatological diseases, treatment with anti-TNF-alpha induces a small reduction in depression (−0.24; 95% CI [−0.33 to −0.14]; $p < 0.001$), and anxiety (−0.17; 95% CI [−0.31 to −0.02]; $p = 0.02$, *Abbott et al., 2015*). Also in Inflammatory Bowel Diseases involving Crohn disease (CD) and ulcerative colitis (UC) psychological alterations have shown, due to dietary restrictions, long-term use of medication to control the disease (*Devlen et al., 2014*), an immediate and lifelong psychological impact on daily activities, such as absences from school or work, and difficulties with meeting employment requirements. This leads to a reduction in Health-Related Quality of Life (HRQoL) not only during the relapse phases but also during the periods of remission of the inflammatory disease (*Mancina et al., 2020*).

A recent prospective cross-sectional study has shown that the prevalence of and factors associated with depression and anxiety in patients with rheumatoid arthritis (RA) are common. Functional disability and marital status were significantly associated with increased risk, whereas disease duration of 10 years or more and global health scores were significantly associated with decreased risk of developing anxiety (*Katchmart et al., 2020*).

In dermatological disorder, most of the evidence comes from psoriasis. Psychological comorbidity, suboptimal coping, and low levels of well-being are associated with psoriasis, significantly impacting on patients' lives (*Esaa et al., 2020*).

In recent years various patient outcome measures (PROMs) have been implemented: they define the patient's experience with a disease and its treatment, including impressions, perceptions, and attitudes, in addition, they are measures of the outcome of the disease management reported directly by the patient or, alternatively, by the caregiver (*Bojic, Bodger & Travis, 2017*). Furthermore, PROMs can facilitate the assessment of physical and psychological functioning and identify suboptimal coping strategies. Over the years, several questionnaires and survey tools have been developed, including HRQoL tools, review of opinions and feedback from patients with inflammatory bowel disease (IBD). Evaluation of PROMs in rheumatoid arthritis (RA) was developed in the Outcome Measures in Rheumatology Clinical Trials (OMERACT), leading to the definition of an international standard "core set" used in clinical trials on RA (*Cella et al., 2007*; *Wolfe, Hawley & Wilson, 1996*).

No specific PRO measure is consistently used in the current management of psoriasis. A recent systematic review (*Duvetorp et al., 2019*) revealed several limitations of the existing measures. In 2019, a survey conducted on 22,050 adults affected by psoriasis and psoriatic arthritis skin showed a negative impact on HRQOL, with 16% of patients reported anxiety and depression disorder (*El-Matary, 2014*).

The National Institutes of Health (NIH)'s Patient-Reported Outcomes Measurement Information System (PROMIS®, http://www.nihpromis.gov, *Cella et al., 2007*) is a set of publicly available and standardized Patient-Reported Outcomes (PROs) that cover physical appearance, mental health, and social health. Two recent studies (*Conley et al., 2017*) on 5,296 patients with IBD from the Crohn's and Colitis Foundation of America's Partners Cohort have used PROMIS to evaluate pain, fatigue, sleep disturbances, anxiety, and depression to identify symptom cluster membership among adults with IBD and examine associations between demographic and clinical factors.

In 2015, a large prospective study (*Bartlett et al., 2015*), carried out in the US, provided preliminary evidence of reliability and construct validity of PROMIS to assess RA symptoms and impacts, and feasibility of use in clinical care. PROMIS instruments captured the experiences of RA patients across the broad continuum, especially at low disease activity levels. Finally, a retrospective multi-center study (*Esaa et al., 2020*) conducted on approximately 7,000 patients with dermatological diseases showed that high T scores > 55 measured by PROMIS were significantly correlated with severe disease activity, treatment failure, and uncontrolled disease.

The flexibility and efficiency of the PROMIS relies on an Item Response Theory (IRT) approach (*van der Linden, 2017*). Indeed, IRT methods allow a deeper understanding of the items' relationships to the targeted domain. An IRT-based evaluation of items' psychometric properties allows the investigators to create short-form scales or even computerized adaptive tests for more targeted patient assessment (*Revicki et al., 2009*).

IRT is "a model-based measurement in which trait level estimates depend on both persons' responses and on the properties of the items that were administered (*Embretson & Reise, 2013*)". IRT models differ in terms of the number of parameters that model the relationship between the latent trait and the (probability of) response.

One of the most straight forward ways to validate an instrument using an IRT approach is to test whether data fit well with a Rasch model (*Rasch, 1980*), namely a model where the probability of responding to an item (or, in polytomous items, to a category within each item) is determined only by the difficulty of that item (and of that category), which is on the same scale as the latent trait. Although the so-called one-parameter logistic models (1PL) within the IRT approach are theoretically distinct from the Rasch model, they are mathematically equivalent, and they both imply some desirable measure that matters in this context. Such a model implies that all the items equally discriminate between respondents with low *vs.* high levels of ability trait, and therefore the total sum score can be considered sufficient statistics of the latent trait level.

On the other hand, in the two-parameter logistic model (2PL), an item discrimination parameter is included in the measurement model. The 2PL model is more flexible and might be useful when developing a new scale, for instance, by discarding the least discriminative items. However, under this model, the items can weigh differently in determining the total score, and therefore it does not retain all the desirable properties that characterize the Rasch model. Comparing the fit of a 1PL model to a 2PL model serves as a way to test the hypothesis that the data fits well enough to the Rasch model. If so, we can assume that the custom four-items from the Italian standard Short Form

Anxiety 8a can provide a sufficient statistic for measuring anxiety levels in the Italian context. On the other hand, the PROMIS was validated using a 2PL approach, therefore a better fit for this model would be in accordance with the validation of this instrument, although this would lead to a slightly different interpretation of the total score.

Another core feature that distinguishes this approach to measurement from the classical test theory (*Remmers, 1951*; *Spearman, 1907*, *1913*), is that the test reliability (or information, in IRT parlance) differs between different levels of the latent trait. In fact, the relationship between information and standard error (SE) is defined by the formula $SE(\theta) = 1/\sqrt{I(\theta)}$, where θ is estimated latent trait level, SE is the standard error of θ, and I is information. Such a feature might be highly desirable when choosing items aimed at populations that should be either very high or very low in certain latent traits.

Finally, another compelling reason for using an IRT approach is because the PROMIS was developed using this approach (*Cella et al., 2010*). Indeed, through this approach it is possible to use a small number of items without sacrificing the reliability of the instrument (*Embretson & Reise, 2013*, Chapter 2). In this study, we used IRT analyses to test the psychometric properties of the original anxiety items of the PROMIS.

## MATERIALS & METHODS

### Study cohort

Between October 2018 and October 2019, 286 consecutive patients were enrolled in Gastroenterology, Rheumatology and Dermatology Departments of the University Hospital "Magna Graecia" University of Catanzaro. In addition, 280 healthy controls, carers of hospital patients without cardiovascular, nor dysmetabolic disease were recruited. All patients were over the age of 18 and had signed informed consent. At the time of enrollment, study participants were invited by specialist nurses to complete questionnaires on anxiety through PROMIS, as described below.

Demographic and anthropometric characteristics, smoking use, physical activity, marital status, level of education, and job activity were recorded for the whole cohort of patients and healthy controls.

Patients from each department with an established clinical, endoscopic, radiological, and histological diagnosis underwent a full evaluation of disease characteristics: disease duration, Erythrocyte Sedimentation Rate (ESR), Reactive C Protein (RCP) and previous surgery. Intake of the drugs was collected in each patient. Assumption of not Steroidal Anti-Inflammatory Drugs (NSAID), Mesalamine, Steroids, Disease-modifying antirheumatic drugs (DMARDs) such as Methotrexate and azathioprine and biological therapy was recorded for each patient. Harvey Bradshaw index (HBI; *Harvey & Bradshaw, 1980*) for CD and Mayo Score (MS; *Rutgeerts et al., 2005*) for UC, respectively have been considered for disease activity. Patients were defined in remission when HBI < 5 for CD or MS < 2 for UC. Patients with rheumatological diseases were divided into five groups: Axial Arthritis, Peripheral Arthritis, Systemic Sclerosis, Systemic Lupus Erythematosus (SLE), and Vasculitis. Ankylosing Spondylitis Disease Activity Score (ASDAS; *Lukas et al., 2009*) index was used for axial arthritis: ASDAS > 1.3 indicates active disease;

Disease Activity Score-28 (DAS-28; *Aletaha et al., 2005*) was used for peripheral arthritis: DAS 28 > 2.6 indicates active disease; Medsger Severity Index (*Medsger et al., 2003*) was used for Systemic Sclerosis and Selena-Systemic Lupus Erythematosus Disease Activity Index (SELENA-SLEDAI; *Petri, 2007*) to assess the degree of activity of SLE: Medsger Severity Index > 1 and SELENA-SLEDAI > 5 indicate active disease. Birmingham Vasculitis Activity Score (*Mukhtyar et al., 2009*) to evaluate vasculitis: score > 1 indicates active disease.

Psoriasis Activity Score Index (PASI; *Walsh et al., 2018*) and Eczema Area Severity Index (EASI; *Simpson et al., 2016*) has been used to calculate disease activity in psoriasis and atopic dermatitis, respectively. In both cases, the disease is defined as active if the scores are >0.

## Instruments

**We selected 4 items from the Italian standard Short Form Anxiety 8a, a previously approved translated version.** In this instrument, participants had to rate how often they experienced each of the symptoms (*e.g.*, "I felt nervous") as occurring over the past 7 days on a five-points Likert response format ranging from 0 (*Never*) to 5 = (*Always*).

## Data analysis

### Unidimensionality

To assess the unidimensionality assumption, we pursued a two-fold strategy.

Firstly, we tested whether the unidimensionality assumption for each sub-scale was met. To do so, we pursued a two-fold strategy by conducting a confirmatory factor analysis (CFA) using the *lavaan* library (*Rosseel, 2012*) in R. Given the ordinal nature of the data, we used the Diagonally Weighted Least Squares (DWLS) robust estimator. We evaluated the model fit considering the Comparative Fit Index (CFI), Tucker–Lewis Index (TLI), Root-Mean-Square Error of Approximation (RMSEA), Standardized Root Mean Square Residual (SRMR), and Weighted Root Mean Square Residual (WRMR). We considered cut-off values as adequate if CFI and TLI were > 0.90, RMSEA less than 0.06 (*Hu & Bentler, 1999*), and WRMR less than 1.0 (*Muthén & Muthén, 2010*).

Secondly, we conducted an Item Response Analysis by Exploratory Factor Analysis of polychoric correlations (extraction method = "minres") across the items in each subscale. Then, we examined the ordered eigenvalues from the item correlation matrix (*May, 1993*). When the unidimensionality assumption is tenable for a subscale, the first eigenvalue should be considerably larger than the remaining eigenvalues. To this purpose, we used the *fa.irt* function provided by the *psych* library (*Revelle, 2021*) in R (*R Core Team, 2021*).

We evaluated internal consistency through ordinal Cronbach's α computed from the polychoric correlation and through McDonald's ω computed from the unidimensional CFA model.

### IRT analysis

Similarly to *Fraley, Waller & Brennan (2000)*, we used a Graded Response Model (GRM, *Samejima, 1968*), an IRT model for polytomous categorical ordered responses (such as

Likert-type responses). GRM is a so-called "indirect" item response theory (IRT) model that extends the 2PL for dichotomous item responses. As described in *Fraley, Waller & Brennan (2000)*, within the GRM framework, an item response scale is conceptualized as a series of $k - 1$ response dichotomies, where $k$ represents the number of response options for a given item. Samejima's model considers the probability of endorsing each response option category, or higher as a function of a latent trait. The response option difficulty represents the point on the latent trait continuum where there is a 50% chance of endorsing the or higher response option. In this sense, the response option difficulty represents a between-option "threshold" parameter. For each item, the number of threshold values equals the number of response dichotomies ($k - 1$). In the 2PL GRM, each item has a single discrimination ($\alpha$) value for all response options. However, the discrimination parameter can be constrained to be equal across items, so to reach some of the desirable mathematical properties of the Rasch model.

GRM models were fitted and analyzed through the *mirt* package (*Chalmers, 2012*), which fits the models using Marginal Maximum Likelihood Estimation (MMLE). We fit two models, one with the discrimination constrained to be equal, and the other one without that constraint, namely allowing different slopes for each item. We compared the two models using a Likelihood Ratio test, but we also looked at the Bayesian information criterion (*Schwarz, 1978*), which favors parsimonious models over complex ones.

The goodness of fit was tested through the M2* statistics proposed by *Cai & Hansen (2013)*, and by looking at absolute (Root Mean Square of Error Approximation, RMSEA; Standardized Root Mean Square Residual, SRMSR) and comparative (Comparative Fit Index, CFI; Tucker-Lewis Index, TLI) indices. Absolute indices should be the closest to zero as possible, but a good fit is achieved with values < 0.05, whereas comparative fit indices should approach the value of one, and they are considered good if > 0.95 (*Hu & Bentler, 1999*).

We tested the assumption of local independence by using the Q3 statistics (*Yen, 1984*), which checks whether two items are correlated after controlling for the latent trait level. These correlations are deemed as problematic if their absolute value exceeds 0.2 (*Yen, 1993*).

We assessed the differential item functioning (DIF) using the ordinal logistic regression method (*Zumbo, 1999*). This method is quite straightforward, as it tests for whether adding the effect of the group, and its interaction with theta scores significantly changes the explained variance of a model in which theta scores were entered first. Through this method, a significant effect of the group is a sign of uniform DIF, whereas a significant interaction between theta scores and the group is a sign of non-uniform DIF. On top of that, we assessed also the effect size of the DIF by checking the McFadden pseudo-$R^2$ (*Menard, 2000*) change. To this purpose DIF analyses were conducted using the *lordif* (*Choi et al., 2016*) package in R, as kindly suggested by a Reviewer.

We assessed DIF as a function of demographic variables such as gender and age (65 years *vs.* 65 years or older (*Pilkonis et al., 2011*)), and for the groups of patients (patients with inflammatory bowel disease, IBD; dermatological patients, DER, and rheumatological patients, RHE) *vs.* the Control group (CTR).

*Group differences*

We computed the factor scores (estimated latent trait) levels through the expected *a posteriori* method (EAP) from the 2PL model. To test whether the groups (CTR, IBD, DER and RHE) differed in anxiety levels, we conducted an ANOVA on the factor scores, controlling for gender and age. We followed-up the omnibus analysis through a set of six post-hoc pairwise comparisons (Bonferroni-corrected). Furthermore, we computed the T-scores on the basis of the parameters estimated in the PROMIS normative data (see Table 4), computed through the Health Measures Scoring Service (https://www.assessmentcenter.net/ac_scoringservice) and conducted the same analyses as for the factor scores. Briefly, we inserted our short form by selecting from the *PROMIS Item Bank V1.0-Anxiety*, we selected the calibration sample *PROMIS Wave 1* and upload our dataset using the template provided.

## Ethics

The Calabria Region regional ethics committee approved this study on 15 March 2018 (protocol number 69). All patients participated in this study in confirmation with the principles outlined in the Declaration of Helsinki. We obtained informed written consent from each participant.

## Data availability

Data and R scripts are available on the Open Science Framework platform at this link: https://osf.io/92ykx/.

## RESULTS

### Descriptive statistics

First, we identified one observation in which an anomalous response was recorded (*i.e.*, responses above five on a five points Likert-type response format), and coded it as missing data. We show demographic and anthropometric characteristics of patients stratified by disease and controls in Table 1. Briefly, out of a total of 566, 286 were patients, and 280 were the healthy controls. Two hundred ninety-one (51%) were male, with a median age of $49 \pm 16$ and a median BMI $= 25 \pm 4$ kg/m$^2$. One hundred ninety-eight (35%) carried out physical activity. Only 100 (18%) was a graduate, 211 (38%) had a job, and 179 (32%) were married.

We show disease characteristics in Table 2. One hundred thirty-six (47%) were suffering from IBD, 100 (35%) were suffering from rheumatic diseases, and 42 (15%) had dermatological diseases. The mean duration of disease was $10 \pm 8$ years, and approximately 29% (80) had active disease. One hundred eighty-three (64%) received mesalazine, 53 (19%) received steroids, and, finally, 95 (23%) received biologic therapy.

A visual check on the response distributions in the four items shows that most of them display a positively skewed distribution because most of the respondents refer to have never experienced anxiety-related symptoms (see Fig. 1). Such a pattern was expected and, in fact, parallels the findings from the original item bank construction study.

**Table 1 Characteristics of patients stratified by study group.**

|  | CTR | IBD | RHE | DER |
|---|---|---|---|---|
| **N** | 280 | 136 | 100 | 42 |
| **Demographic and anthropometric** | | | | |
| Male gender *n* (%) | 162 (58) | 85 (63) | 25 (25) | 19 (47) |
| BMI (Kg/m$^2$) | 25 ± 3 | 25 ± 4 | 26 ± 4 | 24 ± 3 |
| Age (years) | 47 ± 18 | 49 ± 13 | 57 ± 12 | 51 ± 15 |
| Smoke Y *n* (%) | 124 (44) | 27 (20) | 49 (49) | 10 (25) |
| Phisical Activity Y *n* ( %) | 104 (37) | 53 (39) | 26 (26) | 15 (38) |
| **Education level *n* (%)** | | | | |
| Diploma | 177 (63) | 93 (69) | 49 (49) | 20 (50) |
| Degree | 61 (22) | 16 (12) | 18 (18) | 5 (12) |
| **Marital status *n* (%)** | | | | |
| Single | 117 (42) | 29 (22) | 23 (22) | 11 (28) |
| Married | 146 (52) | 107 (78) | 78 (78) | 27 (67) |
| **Occupation *n* (%)** | | | | |
| Employee | 100 (35) | 72 (53) | 27 (27) | 12 (28) |

**Note:**

Continuous variables are shown as mean ± standard deviation, categorical variables are presented as number and proportion. Abbreviations: BMI, Body Mass Index; CRT, Control; IBD, Inflammatory Bowel Disease; Rheuma, Rheumatological Disorders; Derma, Dermatological Disorders.

## Dimensionality

The confirmatory factor analysis on the unidimensional model showed a good fit in terms of CFI and TLI (both equal to one), SRMR (.007) RMSEA (0), and WRMR (0.24).

Results from the EFA on the polychoric correlations show a dominant first dimension (see Fig. 2), and a one-factor solution is well justified by both a scree test approach and a *mineigen* approach (*eigenvalues* > 1). Overall, taking the EFA and the CFA analyses together, the unidimensional solution seems well-justified. Therefore, we decided to pursue the following IRT analyses assuming unidimensionality (see Fig. 2). The items showed excellent reliability both in terms of polychoric Cronbach's α computed from the polychoric (0.93) correlation and through McDonald's ω computed from the unidimensional CFA model (0.92).

## Graded response models

The Likelihood ratio test showed a significant improvement with the unconstrained model (2PL), even in terms of BIC (see Table 3). Moreover, we conducted a M2* test (*Cai & Hansen, 2013*, C2 variant from *Cai & Monroe (2014)*) to test the goodness of fit of our model, which did not reject the 2PL model (4.57, df = 2, *p* = 0.1), all the other fit indices were good (RMSEA = 0.05, SRMR = 0.02, CFI = 1, TLI = 1). The reason why the 2PL model fits the data better, is that there is a noticeable variability in the item discrimination parameters, with the first item ("I felt uneasy") having the lowest discrimination (alfa = 2.58) and the third item ("I felt anxious") having the highest (alfa = 5.95). An inspection of the factor loadings computed from the slope parameters showed that all the items had excellent loadings (>0.7). A visual inspection of the Item

| Table 2 Disease characteristics. | |
|---|---|
| | **N 278** |
| **Inflammatory bowel disease** | |
| Crohn's disease, *n* (%) | 43 (15) |
| HBI | 2 ± 3 |
| Ulcerative Colitis, *n* (%) | 93 (32) |
| MS | 2 ± 2 |
| **Rheumatological disorders** | |
| Axial Arthritis | 9 (3) |
| ASDAS | 1.5 ± 0.9 |
| Peripheral Arthritis | 60 (20) |
| DAS-28 | 2.2 ± 1 |
| Systemic Sclerosis | 14 (5) |
| MSI | 3 ± 3 |
| Systemic Lupus Erythematosus | 10 (3) |
| SELENA-SLEDAI | 3.5 ± 5 |
| Vasculitis | 7 (2) |
| BVAS irmingham Vasculitis Activity Score | 1.2 ± 2.2 |
| **Dermatological disorders** | |
| Psoriasis | 27 (9) |
| PASI | 1.7 ± 1.8 |
| Atopic dermatitis | 15 (5) |
| EASI | 27 ± 1 |
| Disease duration (years) | 10 ± 8 |
| ESR (mm/h) | 14 ± 14 |
| RCP (mg/l) | 6 ± 11 |
| Active Disease n (%) | 80 (29) |
| Surgery, *n* (%) | 17 (6) |
| **Medications, *n* (%)** | |
| DMARDs | 51 (18) |
| FANS | 7 (2) |
| Mesalamine | 183 (64) |
| Steroids | 53 (19) |
| Biological | 95 (23) |

**Note:**

Continuous variables are shown as mean ± standard deviation, categorical variables are presented as number and proportion. Abbreviations: HBI, Harvey Bradshaw Index; MS, Mayo Score; ASDAS, Ankylosing Spondylitis Disease Activity Score; DAS-28, Disease Activity Score -28; MSI, Medsger Severity Index; SELENA-SLEDAI, Selena-Systemic Lupus Erythematosus Disease Activity Index; BVAS, Birmingham Vasculitis Activity Score; PASI, Psoriasis Activity Score Index; EASI, Eczema Area Severity Index; ESR, Erythrocyte Sedimentation Rate; RCP, Reactive C Protein; DMARDs, Disease modifying antirheumatic drugs.

Response Category Characteristic Curves shows that the assumption of monotonicity was met (see Fig. 3).

The single Item Information Curves and the overall test reliability curve (Fig. 4) computed from the test information curve showed that the scale is most reliable for values

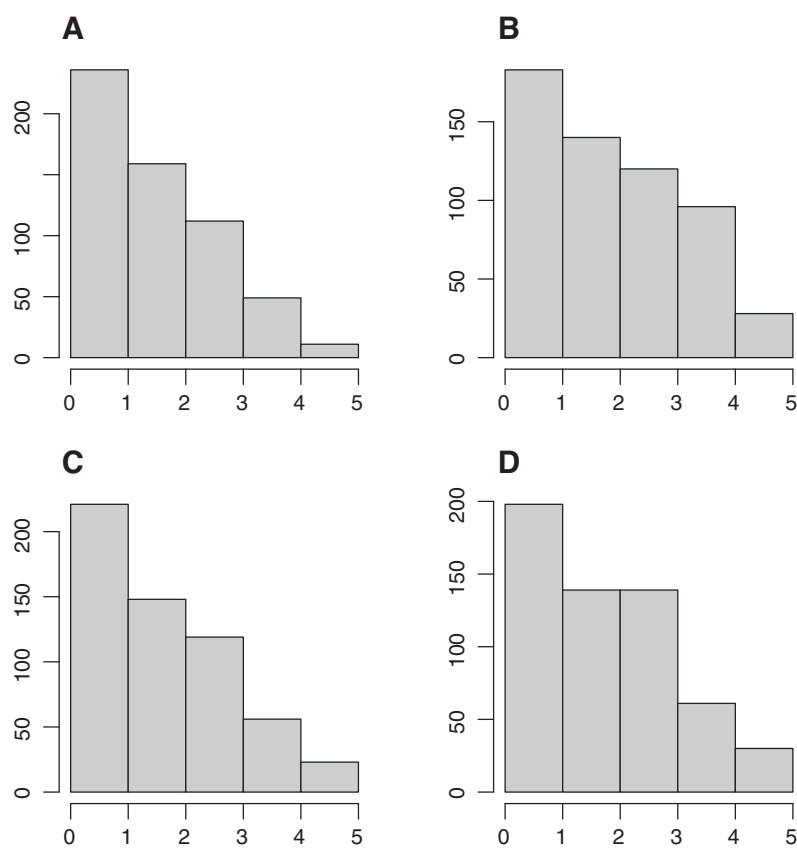

**Figure 1 Response frequency distribution.** Response frequency distribution across items of the custom four-items Italian adaptation from the PROMIS Anxiety 8a short form. A–D show items 1–4 of the custom four-items anxiety 165 short from the PROMIS anxiety 8a short.

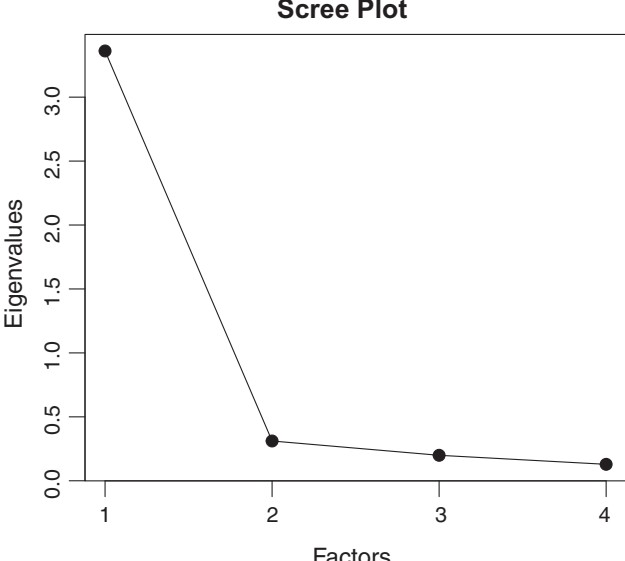

**Figure 2 The scree plot plots the *i*-th factor against its *eigenvalue*.**

**Table 3 Model comparison between the unconstrained (2PL) and constrained (1PL) graded response models.**

| Model | BIC | logLik | Δχ² | df | p value |
|---|---|---|---|---|---|
| Constrained (1PL) | 5,080 | −2,486 | | | |
| Unconstrained (2PL) | 5,045 | −2,459 | 54 | 3 | <0.001 |

Note:
Abbreviations: BIC, Bayesian Information Criterion; logLik, Log Likelihood; LRT, Likelihood Ratio Test; DF, degrees of freedom.

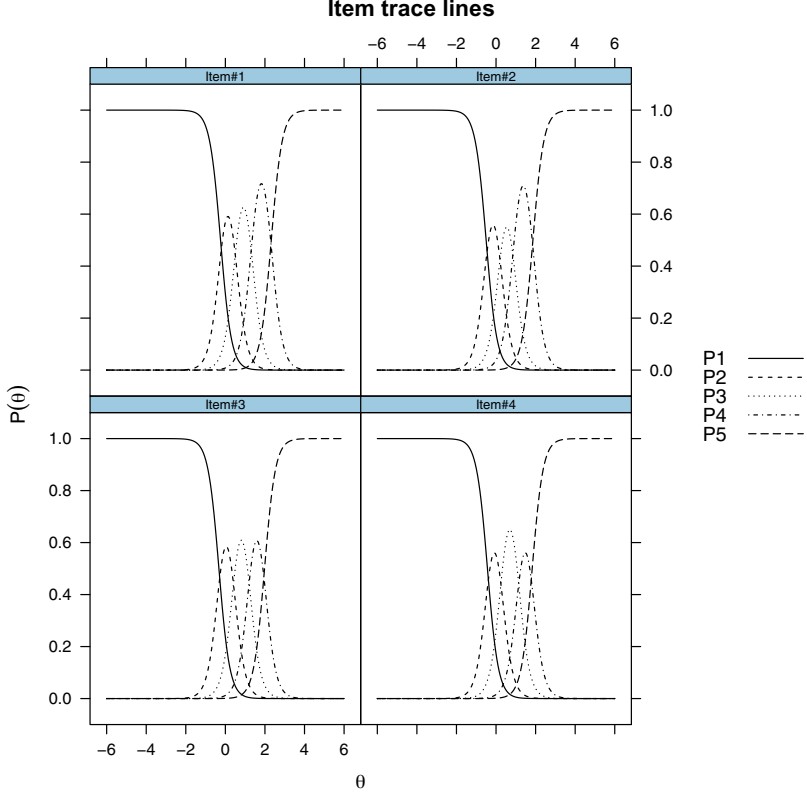

**Figure 3 Item response category characteristic curves of the custom four-items Italian adaptation from the PROMIS Anxiety 8a short form.** Each line represents one response category on the five-points Likert-type response. The y-axis represents the probability of choosing that response category. The x-axis represents different levels of the latent trait (Theta).

of the latent trait that are just below zero up to two logits, meaning that the scale is most reliable for higher levels of anxiety.

The inspection of the Q3 residuals showed that some pairs of items have an absolute value higher than 0.2. This result suggests that local independence conditional on the latent trait seems to be violated, especially for some items. However, most of these correlations were negative, and this might be due to an artifact because the instrument is short (*De Ayala, 2008*). When looking at the positive Q3 values, we found that none of the items were problematic.

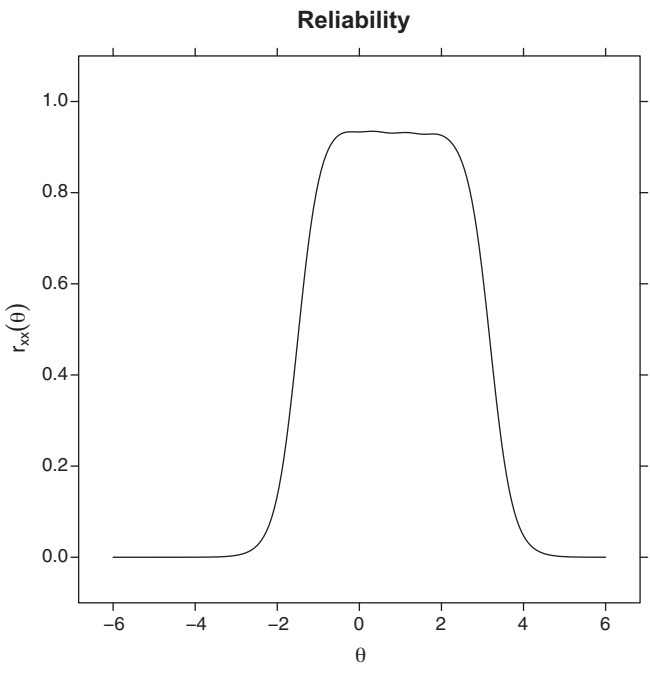

**Reliability**

**Figure 4  Test reliability plot, computed from the test information plot.** Levels of reliability (y-axis) as a function of the estimated latent trait level (Theta).     

**Table 4  T-scores stratified for study group.**

| Group | n | mean | sd | median | min | max | range | skew | kurtosis |
|-------|-----|-------|-------|--------|------|------|-------|-------|----------|
| CTR | 280 | 48.33 | 8.22 | 47.2 | 37.3 | 76.1 | 38.8 | 0.34 | −0.36 |
| IBD | 144 | 54.21 | 11.97 | 53.35 | 37.3 | 79.9 | 42.6 | 0.20 | −0.86 |
| RHE | 100 | 55.77 | 9.92 | 57.1 | 37.3 | 79.9 | 42.6 | −0.23 | −0.52 |
| DER | 43 | 54.93 | 12.54 | 57.1 | 37.3 | 79.9 | 42.6 | −0.03 | −1.08 |

**Notes:**
Abbreviations: CTR, Control group; IBD, Inflammatory Bowel Disease; RHEUMA, Rheumatological disorder group; DERMA, Dermatological disorder group.
T-scores were computed through the Health Measures Scoring Service (https://www.assessmentcenter.net/ac_scoringservice) on the basis of the parameters estimated in the PROMIS normative data.

## Differential item functioning

In this analysis, we set the alpha value as $0.05/4 = 0.0125$ to correct for the Family-Wise Error Rate.

*Gender* We found no evidence for either uniform or non-uniform DIF ($ps \geq 0.0125$)

*Age.* We found no evidence for either uniform or non-uniform DIF ($ps \geq 0.07$).

*Groups of patients* vs. *Control group.* We found evidence ($p < 0.0125$) for a uniform and non-uniform DIF in for all the items. However, when looking at the effect size for such DIF effects, they were all small (MacFadden $\Delta Rs^2 < 0.13$).

## Group differences in anxiety levels

After computing the factor scores (estimated latent traits), we ascertained that the factor scores and the total sum scores displayed a very high correlation coefficient (Pearson's

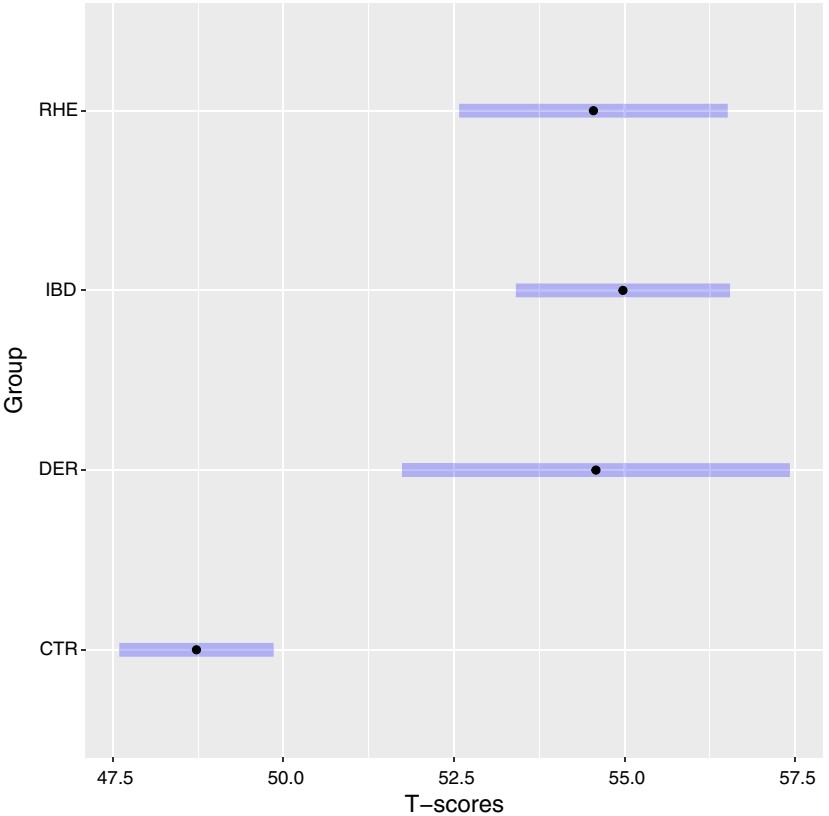

**Figure 5** **Least square means and 95% confidence intervals of the T-scores of each group, controlling for age and gender.** T-scores were computed through the Health Measures Scoring Service (https://www. assessmentcenter.net/ac_scoringservice) on the basis of the parameters estimated in the PROMIS normative data. Abbreviations: CTR, Control group; DER, Dermatological disorder group; IBD, Inflammatory Bowel Disease group; RHE, Rheumatological disorder group.

$r = 0.99$). However, it is worth noting that, because the data do not conform to a Rasch model, the same raw scores could reflect slightly different levels of latent traits and, therefore, of factor scores. We found a significant effect of the group ($F(3, 561) = 19.3$, $p < 0.001$) on factor scores.

The post-hoc comparison showed that the Control group displayed significantly lower levels of anxiety levels ($p$s $< 0.001$, Bonferroni-corrected) compared to all the other patent groups. On the other hand, the patient groups did not significantly differ each from the other ($p$s $= 1$, Bonferroni-corrected). Running the same analyses on the raw scores led to the same results. We conducted the same analyses on the T-scores, which were nearly perfectly correlated with the factor scores computed locally ($r = 0.99$). We found a significant effect of the group ($F(3, 561) = 18.57$, $p < 0.001$) on T-scores (see Fig. 5).

The post-hoc comparison showed that the Control group displayed significantly lower levels of anxiety levels ($p$s $< 0.001$, Bonferroni-corrected) compared to all the other patent groups. On the other hand, the patient groups did not significantly differ each from the other ($p$s $= 1$, Bonferroni-corrected).

## DISCUSSION

In agreement with the Promis Health Organization, we tested for the first time a PROMIS anxiety custom four-item short form in Italian adapted from the PROMIS Anxiety short form 8a. The use of patients reported outcomes to identify anxiety has numerous advantages in the context of chronic inflammatory diseases such as IBD, rheumatic and dermatological diseases. Measuring health outcomes is of pivotal importance in clinical practice. Quantifying in a standardized way is less time consuming and does not require trained personnel. The PROMIS database consists of a large items bank that has been extensively validated using an IRT approach (*Cella et al., 2010*) with the aim of goal creating and evaluating a set of publicly available, flexible, and efficient measurements of important symptoms and patient function that could be easily be used by the clinical research community.

In this study, we selected 4 items from the Italian standard Short Form Anxiety 8a and administered them an Italian sample ($n$ = 568, 271 F) that included Dermatological, Rheumatological patients, and IBD patients, using an Item Response Theory approach. Such a sample size is above the minimum recommended ($n$ = 500) for fitting 2PL models (*De Ayala, 2008*).

First, we confirmed the unidimensionality of the measurement, using a CFA for categorical ordered variables in conjunction with an EFA on the polychoric correlation. This finding implies that these four items should reflect a specific construct, and this a precondition for evaluating the reliability using a unique estimate and for testing the psychometric properties of the items using an IRT approach.

Secondly, we conducted an IRT analysis for polytomous items using a modified version of Samejima's GRM. However, in a first step, we constrained the item discrimination to be equal in all the items. In the second step, we allowed for a free estimation of the discrimination. We found that the second improved the goodness of fit, the $M2^*$ test confirmed that a 2PL model could not be rejected. Such a finding suggests that our data are not compatible with a Rasch model, and therefore the unweighted sum score cannot be considered a sufficient statistic of participants' latent traits. This is line with what was found in the development of the PROMIS, in which the developers explicitly preferred the greater flexibility provided by a 2PL model as compared to a 1PL model. Moreover, we found that, in practice, factor scores (estimated latent trait levels) and raw scores were almost perfectly correlated.

Thirdly, we assessed the reliability using Cronbach'α and McDonald's ω, which both showed excellent reliability in terms of internal consistency from a CTT standpoint. Moreover, our analysis of the test information showed that the test is most informative for higher levels of the latent trait. Overall, our results showed that this custom Italian four-items short form not only reliability estimates anxiety levels but performs better with people who are higher in anxiety. Therefore, the test appears to be especially suited for testing clinical populations such as IBD, Dermatological, and Rheumatism patients.

Fourthly, the inspection of the assumption of local independence further corroborated the validity of this short measure.

Finally, the lack of DIF as a function of gender and age suggests that this shorter measure raw scores could be used to compare these groups. On the other hand, patients and controls exhibited uniform DIF in all the items, although the magnitude of this effect was small.

Overall, these results suggest that this short custom four-item short form in Italian could easily be used.

Coherently with previous data (*Whitehouse et al., 2019*) that showed that IMIDs share an increased prevalence of mood disorders such as anxiety and depression, when we compared the factor scores of the Control group with the other patients' groups, we found significantly lower levels of anxiety in the Control group, as compared to the patients' groups. However, patients affected by different disorders (either Dermatological, Rheumatological or Gastrointestinal) did not differ from each other.

A possible limitation is that in this sample the patients' groups were oversampled in comparison to the control group. One of the advantages of the IRT approach is that item parameters estimation is independent from population assumptions, and that a representative sample is not necessary during the calibration (*Embretson & Reise, 2000*; *De Mars, 2010*). However, it has been shown that, depending on different scenarios, oversampling of patients' population can bias parameter estimation in cases in which items show quasi-trait characteristics, namely when they are poorly informative in the low end of the distribution (*Smits, 2016*). On the effect of adding clinical samples to validation studies of patient-reported outcome item banks: a simulation study. Nevertheless, it is worth noting that, although this may represent a problem when the sample is used for building norms, that was not the goal of the present study. Furthermore, the latent traits (factor scores) estimated in our sample nearly perfectly correlated ($r = 0.99$) with the T-scores computed using the parameters from the US normative sample.

Finally, the DIF analysis showed a less than moderate bias in the items' parameter estimations.

Overall, considering the purpose of the study, and taking empirical and pragmatical consideration into account, the oversampling of the population of the patients should not threaten the validity of our conclusions.

## CONCLUSIONS

In the present study, we selected four items from the Italian standard Short Form Anxiety 8a and administered them to an Italian sample that included both healthy controls and IMIDs patients. We thus validated the anxiety custom four-item PROMIS short form using an IRT approach. The results confirm that these items are well suited for measuring anxiety in clinical populations. Moreover, the use of a four items custom form could prove useful for the brevity and speed of execution in high flow clinics such as those of IMIDs. Further studies are needed to confirm that a shorter items version could be utilized.

## ACKNOWLEDGEMENTS

The authors thank Sabrina Ligarò for data collection and data entry. We also wish to thank Roberto Giorgini for his help in the preparation of the tables.

### Funding

The authors received no funding for this work.

### Competing Interests

Marco Tullio Liuzza is an Academic Editor for PeerJ.

### Author Contributions

- Marco Tullio Liuzza conceived and designed the experiments, analyzed the data, prepared figures and/or tables, authored or reviewed drafts of the paper, and approved the final draft.
- Rocco Spagnuolo conceived and designed the experiments, performed the experiments, analyzed the data, prepared figures and/or tables, authored or reviewed drafts of the paper, and approved the final draft.
- Gabriella Antonucci analyzed the data, authored or reviewed drafts of the paper, and approved the final draft.
- Rosa Daniela Grembiale performed the experiments, authored or reviewed drafts of the paper, supervision, and approved the final draft.
- Cristina Cosco performed the experiments, prepared figures and/or tables, and approved the final draft.
- Francesco Salvatore Iaquinta performed the experiments, authored or reviewed drafts of the paper, and approved the final draft.
- Vanessa Funari performed the experiments, authored or reviewed drafts of the paper, and approved the final draft.
- Stefano Dastoli performed the experiments, authored or reviewed drafts of the paper, and approved the final draft.
- Steven Nistico conceived and designed the experiments, authored or reviewed drafts of the paper, supervision, and approved the final draft.
- Patrizia Doldo conceived and designed the experiments, authored or reviewed drafts of the paper, supervision and Resources, and approved the final draft.

### Human Ethics

The following information was supplied relating to ethical approvals (*i.e.*, approving body and any reference numbers):

The Calabria Region* ethics committee approved this study on 15 March 2018 (protocol number 69).

*Ethics committees are set at a regional level by the Italian law (http://www.eurecnet.org/information/italy.html), and the "Magna Graecia" University of Catanzaro is in Calabria.

## Data Availability

Data and R scripts are available at the Open Science Framework: Liuzza, Marco Tullio, and Rocco Spagnuolo. 2021. "Data and Scripts." OSF. June 15. osf.io/92ykx.

The raw measurements are available in the Supplementary Files.

## Supplemental Information

Supplemental information for this article can be found online at http://dx.doi.org/10.7717/peerj.12100#supplemental-information.

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
