# Peer review of "Psychometric evaluation of an Italian custom 4-item short form of the PROMIS anxiety item bank in immune-mediated inflammatory diseases: an item response theory analysis"

_PeerJ, doi:10.7717/peerj.12100_

## Round 0.1 · original submission · Major Revisions

This article needs a major revision.

In particular, the authors did not use an approved translation, and translation of PROMIS measures is not allowed without prior written permission from the provider and the authors need to follow the PROMIS terms of use.

·

Basic reporting

In general, the paper is well-written, however it would benefit from a careful read-through and editing by a native English speaker (e.g., line 129: "permits to assemble")

It would be nice to see actual anxiety level data shown for the patient versus control groups, and patient subgroups, to illustrate degree of anxiety relative to the general population, and have more text to this effect. Unless i missed it, after reading the manuscript, one has no idea how anxious patients to inflammatory diseases are relative to one another or the general population, or the control sample. This seems odd to me. please provide at minimum the mean T scores and standard deviations around those means. Some of the

Experimental design

The logic of the study is clear, and the methods used are state-of-the-art, with appropriate software documentation. all-in-all, it is clear the 8-item anxiety measure fits a 1PL model which affords it some potential advantages in application. To some extent, the authors get a little lost in this explanation regarding 1PL vs 2PL GRM results, and might consider scaling that back in favor of more information about the nature of anxiety in the inflammatory diseases studied.

Validity of the findings

The authors state that they translated the 8-item anxiety short form, but do not report whether this translation has been recognized and approved by the PROMIS Health Organization (PHO) or Northwestern University, acting on behalf of the PHO. It is also not clear exactly which 8-item version was translated. the specific version name, as specified at http://www.healthmeasures.net, should be provided. in the method section, after which a 'shorthand' referencing may be applied.

I have some concern about this issue in the discussion: "Finally, the inspection of the assumption of local independence suggested that one of the items (either "I felt anxious" or "I felt tense") could be removed as they semantically overlap with other items ("I felt uneasy"). Overall, these results suggest that this item bank could be reduced to six items in future studies, making the instrument even more agile for testing clinical populations." First, the authors did not study a bank; they studied an 8-item short form drawn from a larger bank. Second, the words "anxious" "tense" and "uneasy" are not very overlapping in English. perhaps the Italian translations were more overlapping, causing the observed result. Thus, it is reasonable for them to conclude that the Italian (but not necessarily English) version of this short form could be reduced to 6 items without much loss of information.
a related concern is the return to this finding in the short conclusion. As the authors themselves point out, an item bank affords multiple assessment options, from fixed short forms of varying lengths, to computerized adaptive testing. this paper studies one particular short form, and concludes that (in Italian version) the six-item version may work as well as the 8-item version. I would think that the short conclusion would be borader than this narrow finding; for example stating the multiple sources of support for the Italian PROMIS Anxiety 8-item short form.

Additional comments

Minor comments:
throughout: the ® in PROMIS® need only be used in the title and first use of the term in abstract and body

Line 88: "married" should be "marital"
Line 87 gives "RA" as abbreviation for rheumatoid arthritis, and then line 102 flips it to "AR". use "RA throughout
line 103: 'the definition of a "core set" of eight results' - results is not the right word here
line 111: Conley et al., 2017 is not the correct reference for PROMIS. Use cella references here

·

Basic reporting

The authors have performed a validation study of an Italian translation of a PROMIS Anxiety Short Form.
Unfortunately, the authors did not use an approved translation. Translation of PROMIS measures is not allowed without prior written permission from the provider, see the PROMIS terms of Use
https://www.healthmeasures.net/images/PROMIS/Terms_of_Use_HM_approved_1-12-17_-_Updated_Copyright_Notices.pdf

I recommend the authors to contact the PROMIS language coordinator at translations@HealthMeasures.net to request for approval of the translation. Without approval this paper cannot be published. Note that an official 8-item Italian translation of PROMIS Anxiety does exists. It has to be checked whether there is no overlap between the translations to prevent two different translations of the same items.

Caroline Terwee
President PROMIS Health Organization

Experimental design

I have not further reviewed the paper

Validity of the findings

I ahev not further reviewed the paper

---

## Round 0.2 · Major Revisions

This manuscript can be improved according to the suggestions of Reviewer 2.

·

Basic reporting

I have one major concern about the current paper. It is not clear to me whether the conversion table used for scoring (Appendix 1) was created using the item parameters from the original US calibration sample. This could be done by using the HealthMeasures Scoring Service tool. This should be confirmed because it is not allowed to create T-score tables based on local item parameters. Also, all T-scores for the study population, presented in the paper, should be based on the US item parameters (again, T-scores can be obtained through Scoring Service). See the PROMIS reporting guidelines (Hanmer et al, JPRO 2020;4:21).

Some other comments for further improvement of the paper:

1. The word ‘adaptation’ can be removed from the title because the word ‘custom short form’ is sufficient and common ‘PROMIS language’. It is also not necessary to mention the 8-item short form in the title, referring to the item bank where the 4 items originate from would be more clear. The domain name of the item bank should be presented with capital A (Anxiety). Suggested title: “Psychometric evaluation of an Italian custom 4-item Short Form of the PROMIS Anxiety item bank in immune-mediated inflammatory diseases: an item response theory analysis.”
2. The aim in the abstract should refer to the custom short form, not the original anxiety items. Suggestion: “The aim of this study is to test the psychometric properties of an Italian custom 4-item Short Form of the PROMIS Anxiety item bank in a cohort of outpatients with IMIDs.”
3. The first sentence in the methods section of the abstract should also be rewritten because it still refers to the original 8 items and suggests that 8 items were tested. I suggest to state that 4 items were selected from the standard 8a Short Form. Suggestion: “We selected 4 items from the Italian standard Short Form Anxiety 8a and administered them…..” I suggest to make the same change to line 170-172, 222-223, and 388-389.
4. Line 224-227: the comment about the back translation can be removed. It should be stated that 4 items from a previously approved translated version are being used. Also, line 398 should be corrected as translation of these 4 items was not performed in this study.
5. I wonder if the term “Test Reliability Curve” shouldn’t be “Test Information Curve”. The latter term seems more common.
6. The last sentence of the abstract (“It is advised to compute T-scores based on the estimated latent trait in order to compare different groups”) seems to suggest that are also alternative ways to compute scores (e.g. raw scores), but that is not the case. It is required to present PROMIS scores as IRT-based T-scores, not raw scores (see PROMIS reporting guideline). Please rewrite this sentence. Also lines 426-427 and lines 440-441 seem to suggest that raw scores can be used. These lines should also be rewritten.
7. Line 175: I suggest to change the phrase “change for different levels of the latent trait” into “differ between different levels of the latent trait”
8. Line 132-133: “physical appearance” should be “physical health”
9. Line 169-170 seems to suggest that you will be comparing a Rasch model and a 2PL model in this study, this sentence is a bit confusing. It would be helpful to clarify that PROMIS was developed using a 2PL model and you will therefore also use this mode for testing the 4-item short form.
10. Line 245, the phrase “we conducted an Item Response Analysis by Exploratory Factor Analysis” is confusing because EFA is not an IRT analysis. The IRT analysis is described in the next paragraph. Please remove “Item Response Analysis by” from this sentecnce.

Experimental design

1. It seems that two samples were used for calibration, 286 were patients, and 280 were healthy
controls. It is not clear if it is appropriate to combine two different samples into one study. Smits for example, showed that “ignoring the addition of extra clinical respondents leads to bias in item and person parameters“ (Smits Qual Life Res 2016, DOI 10.1007/s11136-015-1199-9).
2. Methods, line 235 and further. I do not understand why it is stated that unidimensionality and IRT analyses were perform “for each subscale”. Isn’t there only one scale?
3. Why was MAP used for score estimation instead of EAP as used by PROMIS?

Validity of the findings

1. DIF between patients and controls was found for three out of the four items. It would be helpful to present an indication of the magnitude of the DIF (e.g. MacFadden’s R2change) and to analyze the impact of DIF on total scores (this can be done e.g. with the lordif package in R), to see whether this DIF is problematic or not.

Additional comments

I appreciate the efforts that the authors took to collaborate with PHO and Northwestern University to obtain permission for a 4-item custom short form. I am happy to see that the translation issue has been solved and that this 4-item custom short form has now been validated. Since all short forms originate from the same item bank and are assumed to measure the same construct, information on the performance of this 4-item custom short form may also be relevant for the application of other Italian short forms from this item bank. Therefore, the paper is a relevant contribution to the literature.

---

## Round 0.3 · accepted · Accept

The article improved and is now suitable